# ROS Production by a Single Neutrophil Cell and Neutrophil Population upon Bacterial Stimulation

**DOI:** 10.3390/biomedicines11051361

**Published:** 2023-05-04

**Authors:** Svetlana N. Pleskova, Alexander S. Erofeev, Alexander N. Vaneev, Petr V. Gorelkin, Sergey Z. Bobyk, Vasilii S. Kolmogorov, Nikolay A. Bezrukov, Ekaterina V. Lazarenko

**Affiliations:** 1Laboratory of Scanning Probe Microscopy, Lobachevsky State University of Nizhny Novgorod, 603950 Nizhny Novgorod, Russia; 2Department “Nanotechnology and Biotechnology”, R.E. Alekseev Technical State University of Nizhny Novgorod, 603155 Nizhny Novgorod, Russia; 3Laboratory of Biophysics, National University of Science and Technology MISIS, Leninskiy Prospect, 4, 119049 Moscow, Russia; 4Chemistry Department, Lomonosov Moscow State University, Leninskie Gory 1-3, 119991 Moscow, Russia

**Keywords:** neutrophils, ROS production, electrochemical detection, neutrophil heterogeneity, *S. aureus*, *E. coli*, luminol-dependent chemiluminescence, nanoelectrodes

## Abstract

The reactive oxygen species (ROS) production by a single neutrophil after stimulation with *S. aureus* and *E. coli* was estimated by an electrochemical amperometric method with a high time resolution. This showed significant variability in the response of a single neutrophil to bacterial stimulation, from a “silent cell” to a pronounced response manifested by a series of chronoamperometric spikes. The amount of ROS produced by a single neutrophil under the influence of *S. aureus* was 5.5-fold greater than that produced under the influence of *E. coli*. The response of a neutrophil granulocyte population to bacterial stimulation was analyzed using luminol-dependent biochemiluminescence (BCL). The stimulation of neutrophils with *S. aureus*, as compared to stimulation with *E. coli*, caused a total response in terms of ROS production that was seven-fold greater in terms of the integral value of the light sum and 13-fold greater in terms of the maximum peak value. The method of ROS detection at the level of a single cell indicated the functional heterogeneity of the neutrophil population, but the specificity of the cellular response to different pathogens was the same at the cellular and population levels.

## 1. Introduction

The estimation of the reactive oxygen species (ROS)-producing ability of neutrophil populations has been used to explore physiological [1,2] and pathological [3,4] conditions because neutrophil granulocytes (NGs) not only are a central cellular component of the body’s nonspecific resistance system, but also play a regulatory role as a mediator in inflammation processes [5]. Recently, it has been generally accepted that the neutrophil populations are extremely heterogeneous [6], both in terms of receptor expression and in the functions performed [7,8]. According to their functional properties, neutrophils have been divided into anti-oncogenic [9,10] and pro-oncogenic [11,12], as well as anti-inflammatory [13] and pro-inflammatory [14], and a subpopulation of neutrophils with pro-angiogenic properties has been identified [12,15]. In addition, some conditions, such as trauma, significantly increase the heterogeneity of the neutrophil population [16]. The heterogeneity of cells in terms of morphological properties was also noted. They were divided into (1) cells with a well defined granularity of the cytoplasm (about 60%), (2) cells with a moderately pronounced granularity of the cytoplasm (about 30%), and (3) cells with no cytoplasmic granularity (about 10%) [17]. An interesting aspect was the near-complete coincidence of the functional activity of neutrophils in the reactions of the phagocytosis of quantum dots with this morphological gradation. According to the reaction with quantum dots, neutrophils were divided into the following subpopulations: (1) cells with high chemotactic activity, rapidly absorbing quantum dot aggregates, and surrounded by a halo of quantum dots (about 60%); (2) cells that did not show any chemotactic activity, but quickly accumulated quantum dots in their volume (about 29%); and (3) cells with high motor activity in the cytoplasm and not capable of quantum dot phagocytosis (about 11%) [18]. A similar relationship between the morphology of neutrophils and their functional activity was described in [19], where the studied blood neutrophils of mice with different sensitivity to infection were divided into three subpopulations: (1) PMN-N (phenotype CD49d−, CD11b−, TLR-2,-4,-9, with round nuclei), not affecting macrophages; (2) PMN-I (phenotype CD49d+, CD11b−, IL-12, CCL3, TLR-2,-4,-5,-8, with multilobular nuclei), activating M1-macrophages; and (3) PMN-II (phenotype CD49d−, CD11b+, IL-10, CCL2, TLR-2,-4,-7,-9, with ring-shaped nuclei), activating M2-macrophages.

According to the ability to generate ROS (estimated by the reduction reaction of nitroblue tetrazolium (NBT)) in the work of Gerasimov I. G. and Ignatov D. Yu. (2001), neutrophils were divided into two subpopulations: (1) cells that were activated in a spontaneous reaction of NBT (Ns) (about 30%) and (2) cells that were activated only in a stimulated reaction NBT (Nk) (about 70%) [20]. Later, the functional inequivalence of neutrophils was found not only in the blood, in particular, but Puga et al., (2011) also described two subpopulations of splenic neutrophils that had a phenotype different from that of circulating neutrophils and divided them into NBH1 and NBH2 (neutrophil B cell-helper 1 and 2), inducing T-independent B-cell activation and causing class-switching, those with somatic hypermutations, and those inducing the synthesis of immunoglobulins by B cells [7]. In addition, Deniset et al., (2017) found two additional subpopulations of neutrophils that could inhabit the red pulp of the spleen: Ly6Ghi and Ly6Gint [8].

Therefore, while ROSs in neutrophils are the most important bactericidal factors, it is extremely important not only to estimate the total ability to form ROS by the entire population of neutrophils, but also to develop an approach that would evaluate the ability of an individual cell to form ROS, with a high temporal resolution. Neutrophils form ROS as a result of the assembly of NADPH-oxidase, which is a multicomponent electron transport complex; its two subunits (p22phox and gp91phox) form a membrane heterodimeric flavohemoprotein (cytochrome b558). In the absence of cellular activation, the cytosolic components of NADPH-oxidase (p40phox, p47phox, and p67phox) are not associated with cytochrome b, and the oxidase is in an inactive state. After cellular activation, cytosolic components are transferred to the membrane and associated with cytochrome b, resulting in the formation of a functional NADPH-oxidase [21].

There are many methods for ROS detection inside cells, such as fluorescence, chemiluminescence, the electron paramagnetic resonance (EPR) method, and electrochemical methods [22]. Despite the numerous advantages of the widely used fluorescent methods, they have several disadvantages, particularly difficulty in measuring the signal from a single cell with high temporal resolution. Additionally, important approaches for studying cell metabolism and the overall cell state include scanning probe microscopy methods, which allow for the evaluation of the cytoskeletal condition and comparison of the effect of ROS on the mechanical properties of cells [23,24].

An important approach in the study of ROS is the usage of electrochemical methods, which can be used to evaluate the production of ROS in a single cell during its activation. Electrochemical sensors allow one to perform real-time measurements with high temporal resolution. In particular, it is possible to detect important biological molecules that are released by the cells, such as neurotransmitters. The use of platinum carbon microelectrode amperometry has previously been demonstrated to enable the detection and quantification of ROS and reactive nitrogen species (RNS) released by lymphocytes [25], human fibroblasts [26], and cancer-prone human fibroblasts [27]. The release of ROS/RNS inside and outside cells’ phagolysosomes under the influence of interferon-γ and lipopolysaccharide was demonstrated by electrochemical methods [28,29,30].

Recently, for the first time, platinized nanoelectrodes have been developed and have been shown to facilitate high-temporal-resolution analysis of intracellular ROS/RNS production. Their high sensitivity made it possible to detect extremely low intracellular ROS concentrations; in particular, ROS inside tumors in mice were measured in real time using the developed method [31].

The aim of this study was to compare the intracellular production of ROS in individual neutrophils upon stimulation with *S. aureus* and *E. coli* with an electrochemical method using a platinized nanoelectrode. Additionally, the collective extracellular ROS production by the entire neutrophil population was evaluated during the same stimulation, using luminol-dependent chemiluminescence.

## 2. Materials and Methods

**Isolation of human blood neutrophils.** Neutrophils were isolated from the venous blood of healthy donors, and they were stabilized with sodium heparin (50 IU/mL) (Ellara, Pokrov, Russia). The blood was provided by Nizhny Novgorod Regional Blood Center. N.Ya. Klimova. The study was approved by the Bioethics Commission of Lobachevsky State University (created on 11 November 2016, order on creation No. 497-OD), protocol No. 9 dated 17 July 2017. Blood was diluted with phosphate-buffered saline (PBS), containing 0.137 M NaCl and 0.0027 M KCl (Vecos, Nizhny Novgorod, Russia), pH 7.35 in a ratio of 1:1, and then it was centrifuged on the double ficoll-trazograph (Dia-m, Moscow Russia; JBCPL, Mumbai, India) gradient (ρ = 1.077 g/mL, ρ = 1.110 g/mL, 400 g, 40 min). Isolated neutrophils were washed twice in PBS (400 g, 3 min). The viability and purity of the fraction were evaluated using flow cytometry. Specifically, cells were treated with 1 µg/mL propidium iodide (Invitrogen, Waltham, MA, USA) then analyzed via BD FACS Calibur Flow Cytometer (Becton Dickinson, Franklin Lakes, NJ, USA). Both indexes were at the level of 98–99%.

**Bacterial culture preparation.** The strains of *S. aureus* 2879 M and *E. coli* 321 were grown (37 °C, 24 h) on GRM agar (FBIS SRCAMB, Obolensk, Russia), and then they were washed off the substrate with sterile PBS. Cells were washed three times (1800 g, 10 min) then suspended in the same medium. The optical density of suspension was adjusted by a spectrophotometer (SPECS SSP 705, Moscow, Russia) to 0.75 for *S. aureus* and 0.85 for *E. coli* (λ = 670 nm), which corresponded to 1 × 10^9^ cells/mL. Only non-opsonized bacteria were used for the experiments.

**Fabrication and calibration of nanoelectrodes.** Platinum nanoelectrodes (PtNEs) were fabricated based on commercially available carbon nanoelectrodes (CNE) from ICAPPIC Limited, London, UK). These CNEs had diameters ranging from 50–150 nm and were prepared using previously described methods [31]. Briefly, the CNEs were initially placed in 1 mM ferrocene methanol in PBS solution to estimate size of electrochemical surface. Nanocavities etched into the carbon electrode were utilized to improve the adhesion of platinum. This was achieved through electrochemical etching using CV from 0 to 2.2 V in 0.1 M KOH and 10 mM KCl for 15–40 cycles until nanocavities were formed. The etching process resulted in two peaks on the voltammogram that corresponded to the complete oxidation of FcMeOH and the reduction of ferrocenium inside the nanocavity. Subsequently, platinum was electrochemically deposited onto the carbon surface of the nanoelectrode to increase its electrochemical activity by cycling from 0 to −0.8 V with a scan rate of 200 mV s^−1^ for four to five cycles in 2 mM H_2_PtCl_6_ solution in 0.1 M hydrochloric acid. It was demonstrated that PtNEs have high reproducibility and exhibit excellent electrochemical performance, with a diameter ranging from 50–150 nm. A more detailed description of sensor fabrication and calibration is provided in [31].

**Study of intracellular ROS production by neutrophil by the amperometric method using platinum nanoelectrodes.** All electrochemical measurements were performed at RT using a two-electrode setup. The reference electrode was silver chloride electrode, specifically a 0.3 mm AgCl-coated Ag wire. All potentials are reported vs. Ag/AgCl reference electrode. The Faradaic current was measured with a MultiClamp 700B patch-clamp amplifier (Molecular Devices, San Jose, CA, USA). The data of measurement were transferred and recorded onto a computer with the use of the ADC-DAC converter Axon Digidata 1440B and software pClamp 10 (Molecular Devices, San Jose, CA, USA). The micromanipulator PatchStar (Scientifica, Uckfield, UK) was used to control the position of the nanosensor. The entire experimental setup was located on the table of an optical inverted microscope (Nikon, Tokyo, Japan). The signals were filtered with 0.5 kHz lowpass filters.

Neutrophil intracellular ROS production was measured at a constant potential of +800 mV vs. Ag/AgCl. A suspension of neutrophils (2 mL, final concentration 1 × 10^6^ cells/mL) in a sterile Petri dish 35 mm (Corning, Corning, NY, USA) was incubated to facilitate spontaneous adhesion (37 °C, 20 min). Then, a nanoelectrode was inserted to one of the adherent neutrophils. Then, the neutrophil phagolysosomes randomly collided with the surface of the Pt nanoelectrode. A bacterial suspension (200 µL, final concentration of bacteria 5 × 10^8^ cells/mL) was then added to the neutrophils in the dish. A change in current was observed. The experiment was repeated three to four times, and the duration varied depending on the cell and its response. However, despite this, the overall time was always close to 25 min. Longer time of the experiment was not possible because the neutrophil has high mobility, and the analysis of the moving cell by a nanoelectrode was not possible.

The recorded chronoamperograms were further processed, and the area under the curve of each peak was calculated, which corresponded to the charge and was measured in coulombs. The larger the area under the curve, the greater the charge, and the more ROS molecules were produced inside the cell. Therefore, in order to compare the action of two types of bacteria, we calculated the average oxidation charge in each cell.

**Study of ROS production by a population of neutrophils by luminol-dependent biochemiluminescence.** To ensure that the cells used produced maximum ROS levels, neutrophils were isolated within 120 min of blood sampling [32]. Experiments were carried out in siliconized dishes to avoid cell priming.

The 0.5 mL of isolated neutrophils (final concentration 1 × 10^6^ cells/mL) were incubated (15 min, 37 °C) to stabilize the fluorescent activity, and then they were mixed in a vial with 0.5 mL of bacteria (final concentration of bacteria 5 × 10^8^ cells/mL) and 100 μL of luminol (AppliChem, Darmstadt, Germany) (final concentration 10^−5^ M). A suspension of neutrophils in PBS with 100 µL luminol was used as a control.

The respiratory burst of neutrophil granulocytes was assessed on a Lumat^3^ LB 9508 chemiluminometer (Berthold Technologies, Bad Wildbad, Germany), which recorded the amount of free radicals formed in relative units. Values were taken discretely once every 6 min for 2 h. The obtained curves were used to determine the main indicators of biochemiluminescence: the time of the peak, the intensity of BCL, and the integral value of the light sum.

**Statistical analysis.** Statistical processing was carried out using the Origin 8.0 program (OriginLab Corporation, Northampton, MA, USA). The boundaries of the normal distribution of quantitative indicators of the samples were determined by the Shapiro-Wilk test. Since the distributions did not meet the normality criteria, the median and 25th percentile were determined. The nonparametric Wilcoxon test was used to compare two samples.

## 3. Results

In the first stage of this study, we carried out electrochemical experiments using a nanoelectrode. The main scheme of the experiment is presented in Figure 1A. ROS/RNS contained in phagolysosomes may spill onto the electrochemically active surface and be oxidized, giving rise to individual amperometric spikes. The main parameters of a spike were the area under the spike and its amplitude. Therefore, the measurements were performed coulometrically, and the area under each transient current represented the complete oxidation of the ROS molecule in each phagolysosome. (Figure 1B,C).

It was shown that unstimulated control cells do not actively produce ROS. However, single spikes were present on the control cell chronoamperogram (without stimulation). The control cell chronoamperogram is presented in Figure 2.

Spike signals were shown on the chronoamperograms, indicating intracellular ROS production by neutrophil granulocytes upon their activation by *E. coli.* The response to stimulation varied significantly both among different neutrophils in one donor (Figure 3) and in cells isolated from different donors (Figure 4).

In Figure 3A, it can be seen that certain cells exhibited a high level of spike formation. The amplitude and frequency of spikes correlated with the intracellular ROS production by neutrophils. Conversely, other cells did not respond to *E. coli* stimulation, as evidenced by either the complete absence of spikes or spikes with extremely low amplitudes (Figure 3B,C).

The spikes exhibited a periodic behavior and occurred at varying time intervals with differing amplitudes. Furthermore, the registration time of the spikes varied in different neutrophils. On average, the first series of spikes began at 8.7 ± 3.3 min.

Significant differences were observed in the response of the neutrophils to *S. aureus* stimulation, as compared to the response to *E. coli*. The spikes were characterized by a more pronounced amplitude. The character of spikes was significantly different: a series of spikes was often observed. In comparison to *E. coli*, the initial series of spikes began at an earlier time. Specifically, the first spikes in response to *E. coli* were detected after 8.7 ± 3.3 min, while the first spikes in response to *S. aureus* occurred after 5.4 ± 4.6 min (Table 1). The registration time for the spikes in neutrophils that were activated by *E. coli* was 9.6 ± 3.0 min. Neutrophils stimulated with *S. aureus* had a spike registration time of 8.4 ± 4.2 min. However, there was no significant difference between the two (*p* > 0.05). The respiratory burst in neutrophils activated by *E. coli* was more extended in time and less intense, while it was shorter and more intense in neutrophils activated by *S. aureus* (Figure 4A–C and Figure 5).

The charge (area of spike) and peak amplitude were measured after the release of ROS from the vesicle (Figure 5). The corresponding distributions demonstrated that the quantity of ROS released from each phagolysosome varied. Notably, the amount of ROS produced in response to *S. aureus* was 5.5 times greater compared to that produced in response to *E. coli*.

Luminol-dependent biochemiluminescence (BCL) was used to evaluate the population response of neutrophil granulocytes to stimulation with bacterial strains. The results of a typical experiment in a series of 15 experiments are shown in Figure 6. The statistical analysis of the integrated light sum, height of peak, and time of onset of the luminescence peak is provided in Table 2.

Obviously, both at the population level and the intracellular level, there were similar dynamics of respiratory burst development after stimulation with *E. coli* and *S. aureus*. The neutrophil population response to *E. coli* was considerably weaker and lasted for a more extended period. This was evident from the flat nature of the BCL curve with a low-level plateau. On the other hand, stimulation by *S. aureus* led to the onset of a high-intensity peak on an asymmetric bell-shaped curve, characterized by a steep front. Due to the flat nature of the curve, there were no statistically significant differences between the height of the peak of the negative control (spontaneous neutrophil chemiluminescence) and the height of the peak of the neutrophil population stimulated with *E. coli* (Table 2). However, there were differences in the total light sum and the time of the onset of the peak. Even after the highest level of ROS release within 1 h, the level of formation persisted at a much higher rate than that produced by *E.coli*-stimulated neutrophils. This trend continued up to an hour after the maximum release of ROS.

In addition, the time of the respiratory burst of the neutrophil population in response to *S. aureus* stimulation was observed much earlier, by 25.5 ± 2.4 min, in contrast to the respiratory response of the population to *E. coli* stimulation, the maximum of which was determined at 57.0 ± 9.3 min. These results correspond with the single cell results because the first spike of ROS production in single cell in response to *E. coli* stimulation was 8.7 ± 3.3 min, and, for *S. aureus*, it was 5.4 ± 4.6 min.

## 4. Discussion

The time of the occurrence of the first spikes measured via the amperometric method using a platinum nanoelectrodes corresponds with previous results. It was previously shown that the phagocytosis of non-opsonized *E. coli* began up to 11 min after the introduction of bacteria to neutrophils, and, after 15 min, the phagocytic index reached 60% [33]. The phagocytosis of non-opsonized *S. aureus* was observed 7 min after incubation with the bacteria and after 15 min the phagocytic index reached 70%. In general, the ROS level of production by neutrophils after stimulation with *S. aureus* was significantly higher than the ROS level after stimulation with *E. coli,* either with the amperometric method (Figure 5) or with atomic force microscopy [33].

A similar pattern was observed in the BCL analysis when comparing the population response to single cell response to *S. aureus* stimulation. The maximum production of intracellular ROS by neutrophils was observed at 10–15 min after the initial response. Since the start of phagocytosis may vary between cells, the 25-minute luminescence peak observed by with the BCL method can be the cumulative result of ROS production in the population.

The maximum ROS production in the neutrophil population in response to *E. coli* is difficult to determine, as the reaction plateaus from 6 min into the experiment and the graph does not show a clear peak maximum (Figure 6B). The differences observed in ROS production were likely due to the nature of ROS production and the possibility that the spikes recorded within the same neutrophil may have been induced by different factors. The following processes is observed while measuring the ROS level in the neutrophil cell: (1) ROS production during the assembly of the NADPH-oxidase complex with the participation of cytb558 during the formation of the phagolysosome by specific granules; (2) intracellular ROS of production without the formation of the phagolysosome as a result of heterotypic fusion of the specific (providing cytb558) and azurophilic granules (providing myeloperoxidase), since ROS perform not only a biocidal function against microorganisms, but also an informational function [34]; (3) ROS production of ROS by the neutrophils’ mitochondria [35].

The observed variation in the timing of peak occurrence between single cells, and the neutrophil population may be attributed to the stimulation of the neutrophil population by bacterial strains. Our findings indicate that signals originating from the assembly of the NADPH-oxidase complex (cytb558 complex (gp91phox/p22phox) with p47phox, p67phox, and rac were detected not only in the phagolysosome, but also in the neutrophil membrane, leading to the release of ROS into the extracellular milieu. Partially, this conclusion was evidenced by the similar nature of the BCL curves for the population of the neutrophils stimulated by *S. aureus* and the PMA-positive controls (Figure 6A,B). PMA is known to primarily induce the extracellular release of ROS [36]. In addition to the extracellular release, PMA induces intracellular ROS formation in non-phagosomal compartments [37,38,39]. The observed differences in the onset time of the respiratory burst peak values between a single cell and a population of neutrophils were most likely due to the integral registration of both extracellular and intracellular ROS generation. Although it is possible that “silent cells” may have contributed to the integration of the population response signal, the effect of “quorum sensing” on enhancing this response cannot be completely excluded.

Therefore, in addition to variations in receptor expression on the membranes of different subpopulations of neutrophils, as well as differences in cellular functional activities, a distinct pattern of intracellular ROS generation was observed at the single-cell level. Notably, some cells were not activated, while others produced ROS with varying degrees of activity and duration. However, in the overall neutrophil population, these intercellular differences were leveled out, while specificity of the cellular response to diverse pathogens was still maintained.

Compared to stimulation with *E. coli*, the luminol-dependent BCL method demonstrated that stimulating neutrophils with *S. aureus* resulted in a significantly higher total response of ROS production, with an integral value of the light sum that was seven times greater and a maximum peak value that was thirteen times greater. Therefore, the respiratory response to *E. coli* was much less intense and more extended over time, as compared to the more powerful and compressed response to *S. aureus*, as in the case of a single neutrophil response to the stimulation of a population of cells. These features were common to the response at the single cell level and at the neutrophils’ population level.

The higher ROS-production activity in neutrophils in response to *S. aureus* compared to *E. coli* may be explained by several factors. Firstly, *S. aureus* induces more pronounced nitrosative stress (iNOS expression/NO content/total nitrite) in neutrophils [40]. This stress may be induced by the presence of lipoteichoic acids (LTA), which activate tyrosine kinases and NF-kB in the signaling pathway, triggering iNOS induction [41]. Secondly, the mechanisms of the primary binding between neutrophils and *S. aureus* or *E. coli* via toll-like receptors (TLR) are different. *S. aureus* directly binds to TLR-2 due to the presence of LTA and peptidoglycans, while *E. coli* has a more complex mechanism of interaction with TLR-4, including binding protein and CD14. Unlike monocytes, neutrophils less fre-quently express CD14, so the interaction is less pronounced [42]. Thirdly, staphylococci have virulence factors, such as phenol-soluble modules (PSM), which activate the human formyl peptide 2 receptor, and this binding may lead to a powerful respiratory burst of neutrophils [43].

These results show that differences between the single neutrophil cell and the neutrophil population must be kept in mind during the study of the inflammation process. ROS production is an important parameter to estimate in the reaction of neutrophils to bacterial strains.

## 5. Conclusions

At the population level, differences in the neutrophils respiratory burst were clearly visible for different species of bacteria: the reaction to *S. aureus* was much more pronounced than to *E. coli*. However, the method of luminol-dependent biochemiluminescence did not allow for the identification of the specific cells that contributed to the intensity of the respiratory burst. Hence, the development of a technique that facilitates the investigation of ROS generation at the level of a single cell is important. The electrochemical nanoelectrode method is currently the only method that allows such measurements: using the electrochemical nanoelectrode method, we showed the different reactions of cells to stimulation by the same strain of microorganisms, from complete “silence” to a pronounced reaction with the production of a series of clearly registered spikes. Moreover, the electrochemical nanoelectrode method allowed the variability in the response of individual neutrophils to different microorganisms to be revealed, and, at the same time, it provided additional information about the functional heterogeneity of the neutrophil population.

## Figures and Tables

**Figure 1 biomedicines-11-01361-f001:**
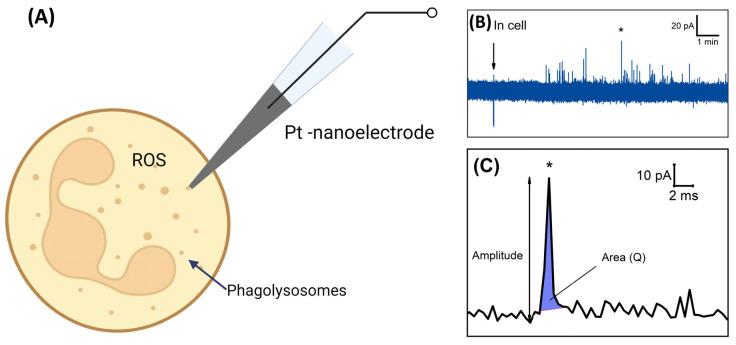
Principle of electrochemical ROS detection inside neutrophils using a PtNE. (**A**) Scheme of the experiment, demonstrating intracellular measurement. (**B**) Chronoamperogram recorded inside the neutrophil. (**C**) Increased spike marked on the chronoamperogram. The asterisk denotes the increased spike in (**C**).

**Figure 2 biomedicines-11-01361-f002:**
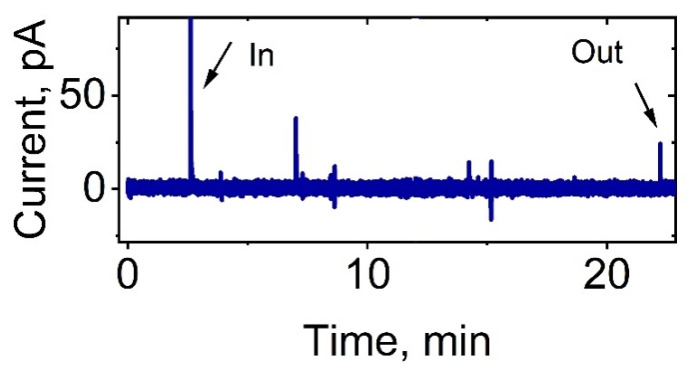
Chronoamperogram recorded at +800 mV vs. Ag/AgCl in intracellular measurements of the ROS level of control neutrophils (without stimulation, negative control).

**Figure 3 biomedicines-11-01361-f003:**
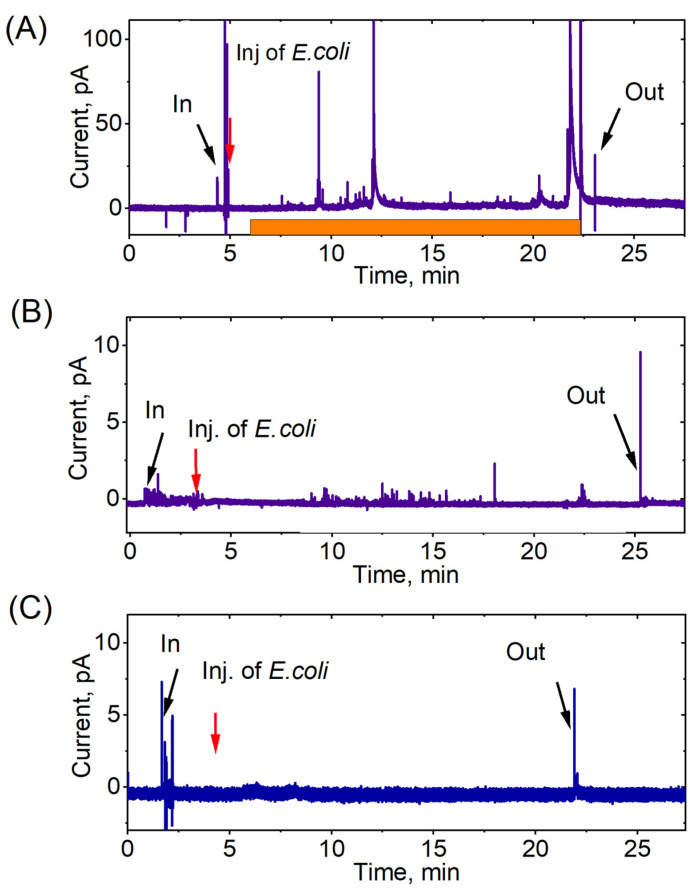
Chronoamperograms recorded at +800 mV vs. Ag/AgCl in intracellular measurements of the ROS level of neutrophils during stimulation with *E. coli*. The time of the first response of a neutrophil after stimulation with a suspension of bacteria is shown. The orange color shows the intervals with active ROS generation. (**A**) Strong reaction of neutrophil to *E. coli*; (**B**) weak reaction of neutrophil to *E. coli*. (**C**) Reaction of neutrophil to bacterial stimulation is absent. Donor S.B. for all experiments.

**Figure 4 biomedicines-11-01361-f004:**
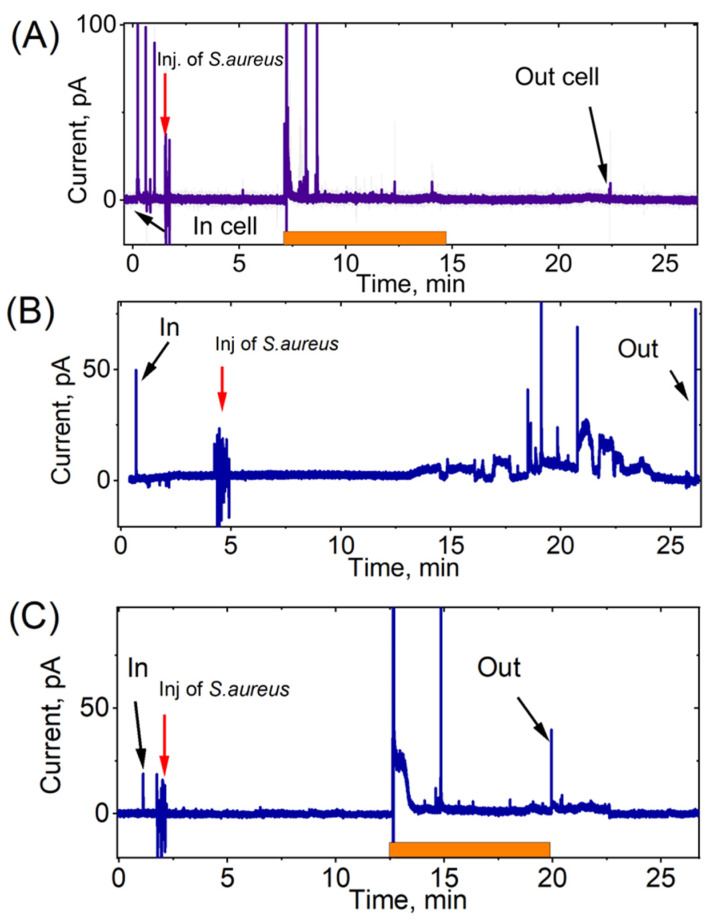
Chronoamperograms recorded at +800 mV vs. Ag/AgCl in intracellular measurements of the ROS levels of neutrophils during stimulation with *S. aureus*. The time of the first response of a neutrophil after stimulation with a suspension of bacteria is shown. The orange color shows the intervals in which the greatest release of ROS occurred. (**A**) Strong reaction of neutrophil to *S. aureus*. Donor S.Z. (**B**) Strong, but delayed, reaction of neutrophil to *S. aureus*. Donor S.P. (**C**) Delayed reaction of neutrophil to *S. aureus.* Donor S.B.

**Figure 5 biomedicines-11-01361-f005:**
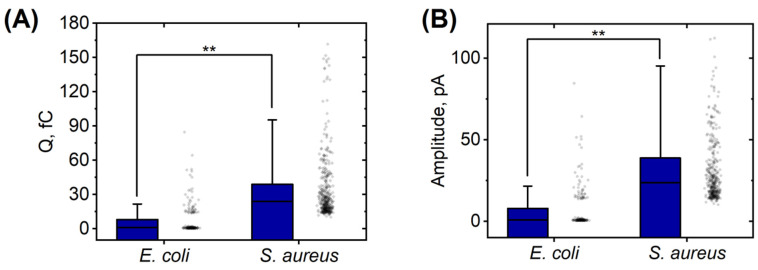
Comparison of ROS production in neutrophils under the influence of two different bacterial cultures (*E. coli* and *S. aureus*). (**A**) Mean area under the spike curve, shown as the charge passed through the oxidation of the phagolysosome. Values are presented in femtocoulombs. (**B**) Average spike amplitude in picoamps. Number of spikes for *E. coli* (N = 166), *S. aureus* (N = 392). Data for *E. coli* are presented for cells taken from the donors S.Z. (two cells), S.P. (one), and S.B (three). Data for *S. aureus* are presented for cells taken from donors S.Z. (four), S.P. (three), and S.B (three). **—*p* < 0.001 (ANOVA).

**Figure 6 biomedicines-11-01361-f006:**
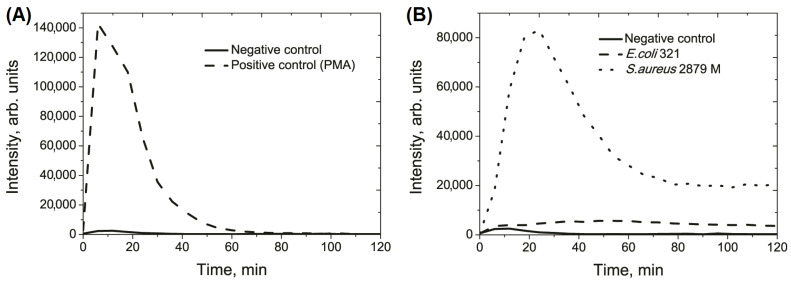
Typical luminol-dependent biochemiluminescnece (BCL) curves. (**A**) Negative control (spontaneous BCL of neutrophils) and positive control (treatment of neutrophils with PMA at a final concentration of 100 ng/mL). (**B**) BCL of neutrophils after stimulation with *E. coli* and *S. aureus*.

**Table 1 biomedicines-11-01361-t001:** Statistics of measured neutrophils.

Strain	Total Cells	Activated Cells	Silent Cells	Average Time of Appearance of Peaks, min	Average Duration of Peaks Appearance, min
*E. coli* 321	6	5	1	8.7 ± 3.3	9.6 ± 3.0
*S. aureus* 2879 M	10	8	2	5.4 ± 4.6	8.4 ± 4.2

**Table 2 biomedicines-11-01361-t002:** The main parameters of luminol-dependent biochemiluminescence of a neutrophil granulocyte population under the influence of *E. coli* or *S. aureus* (*n* = 15).

Parameter	Negative Control	*E. coli* 321	*S. aureus* 2879 M
Integral value of the light sum, arb. units, ×10^5^	1.5 ± 0.4	5.2 ± 1.3 ^a^	39.7 ± 4.9 ^a^
Height of curve peak, arb. units, ×10^3^	5.1 ± 1.7	5.9 ± 1.3	81.1 ± 12.8 ^a^
Time of maximal peak, min	11.5 ± 2.0	57.0 ± 9.3 ^a^	25.5 ± 2.4 ^a^

^a^ The differences between control and experiment are significant (*p* < 0.001).

## Data Availability

The data are available from the corresponding author.

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
