# Peer review of "ROS Production by a Single Neutrophil Cell and Neutrophil Population upon Bacterial Stimulation"

_biomedicines, 2023, doi:10.3390/biomedicines11051361_

Round 1

Reviewer 1 Report

1.       What do the authors mean by single neutrophils?

2.       Abstract-line 19 to 23- very long sentence, please split.

3.       The introduction is too long, please write relation of ROS and neutrophils and raise the question to address the aim of this study (lines up to 41 and then 61 onwards)  followed by another section describing heterogeneity and types of neutrophils (lines 41-60). This should be followed by discussing why there will be different ROS levels with different stimuli (lines 99 onwards).

4.       Methodology has been mentioned repeatedly in results, please check and avoid.

5.       It will be better to separate results and discussion section, as per the journal guidelines. This is important because from line 260 to 319, there are only 2-3 citations and it became difficult to understand if the authors are describing the results or are discussing the results.

6.       Please check the text thoroughly for the grammar (yellow highlights are for example).

7.       The conclusion is supported by the data provided but these is a eed to separately mentioning results and the discussion section.

Improvement needed with editing for grammar and the English language.

Author Response

Dear colleague, thank you very much for your work. We have fixed issues according to your comments.

1.-  What do the authors mean by single neutrophils?

We use this term "single neutrophil" by analogy with the term "single cell". But you probably disagree with the use of the plural. The idea was that we statistically process data from many single cells. We think that you are right, so we corrected it for a more grammatically correct construction.

  1. Abstract-line 19 to 23- very long sentence, please split.

Thank you. We corrected it.

  1. The introduction is too long, please write relation of ROS and neutrophils and raise the question to address the aim of this study (lines up to 41 and then 61 onwards) followed by another section describing heterogeneity and types of neutrophils (lines 41-60). This should be followed by discussing why there will be different ROS levels with different stimuli (lines 99 onwards).

Sorry, but we cannot remove the part that concerns the heterogeneity of the neutrophil population, since it highlights the importance of its variety. Otherwise it becomes unclear why it is so important to move from population studies to the level of single cells. All currently available information on the receptor and functional heterogeneity of neutrophils is specifically listed here. ROS production is only a special case of this heterogeneity.

You are absolutely right what is novelty of the work: for the first time the level of ROS production in single cell was measured.

  1. Methodology has been mentioned repeatedly in results, please check and avoid.

Thank you for your note. We corrected it.

  1. It will be better to separate results and discussion section, as per the journal guidelines. This is important because from line 260 to 319, there are only 2-3 citations and it became difficult to understand if the authors are describing the results or are discussing the results.

Thank you for that suggestion, we revised the structure of the manuscript.

  1. Please check the text thoroughly for the grammar (yellow highlights are for example).

Thank you very much for so detail work with our manuscript and also for your proposed version of the manuscript, because the author's version did not provide for line numbering and thanks to you we understood all the comments made to us when they were "tied" to line numbers. We corrected it.

  1. The conclusion is supported by the data provided but these is a need to separately mentioning results and the discussion section.

Thank you for that suggestion again, we revised the structure of the manuscript.

Reviewer 2 Report

The manuscript titled “ROS production by single cell and populations of neutrophils upon bacterial stimulation” by Pleskova, S.N.; et al. is an original work where the authors address the impact of the presence of two different bacteria (Staphylococcus aureus and Escherichia coli) in the generation of reactive oxygen species (ROS) in neutrophil granulocyte bodies by scanning ion-conductance  microscopy (SICM). The authors found a significantly larger amount of ROS production when the neutrophils were exposed to Staphylococcus aureus in comparison to Escherichia coli.

The most relevant outcomes found by the authors can contribute to better understand the underlying redox mechanisms involved inside of neutrophil granulocytes. This knowledge could have a positive impact to design more efficient anti-inflammatory treatments against diseases caused by bacteria. Furthermore, the methodology followed by the authors can be fully extandable to other immune cells.

Here, there exists some suggestions in order to improve the scientific quality of the manuscript paper:

1) Introduction section is clear and concise. “(…) since neutrophil granulocytes (NGs) (…) play the regulatory role as a conductor in the inflammation process” (lines 34-36). Here, the authors should add some relevant reference citation [1].

[1] Domerecka, W.; et al. Indicator of Inflammation and NETosis-Low-Density Granulocytes as a Biomarker of Autoimmune Hepatitis. J. Clin. Med. 2022, 11, 2174. https://doi.org/10.3390/jcm11082174

2) “the following subpopulations: (…) cells with high chemotactic activity (…) (about 60%), (2) cells that did not show any chemotactic activity (…) (about 28%), and (3) cells with high-motor activity in the cytoplasm and not capable of quantum dot phagocytosis (about 10 %)” (lines 49-54). Why does the sum of all this percentages not render a total of 100%? Please, the authors should fix this point.

3) “(…) neutrophils were divided into two subpopulations: (…) NBT (Nk)” (lines 62-65). Here, the authors should add the percentages according each subpopulation in order to be consistent with the previous information detailed in this section.

4) “The release of ROS/RNS (…) was demonstrated by electrochemical methods” (lines 91-93). Here, the authors show a suitable approach to determine the impact of ROS and RNS inside the phagolysosome cells but it exist many others. For example, the authors should also point out the possibility to use nanoindentation techniques [2] to unravel the effect of ROS/RNS species inside the cell like the recently reported study where the cellular elasticity properties were monitored according the presence of ROS [3]. Finally, the authors should highlight the advantages of those measurements carried out by scanning ion-conductance microscopy like the fact not to touch the sample during scanning which greatly benefits not to damage the cellular samples of interest.

[2] Magazzù, A.; et al. Investigation of Soft Matter Nanomechanics by Atomic Force Microscopy and Optical Tweezers: A Comprehensive Review. Nanomaterials 2023, 13, 963. https://doi.org/10.3390/nano13060963.

[3] Pastrana, H.F.; et al. Evaluation of the elastic Young’s modulus and cytotoxicity variations in fibroblasts exposed to carbon-based nanomaterials. J. Nanobiotechnology. 2019, 17, 32. https://doi.org/10.1186/s12951-019-0460-8.

5) Materials and Methods. Please, the authors should detail all manufacturer details (name and country) for all used chemical agents and consumables. For example, there lacks this information related to employed buffers (PBS, NaCl and KCl in lines 109-110 or GRM agar in line 117). Please, take this comment into account for the rest of the M&M section.

6) (OPTIONAL) Perhaps it may be interesting to add a extra Figure with some results according to the Fabrication and calibration of nanoelectrodes (lines 122-137) as Supplementary Information (SI) in order to see the subsequent oxidation of ferrocenemethanol molecules and platinum deposition in each consecutive cycle.

7) Results and Discussion. Figure 1, panel B (line 192) and Figure 2 (line 197). Are the axes of these chronoamperograms (with and without stimulation respectively) the same range of values? Same comment for the X-axis of Fig. 3, panel C respect the respective panels A and B (line 214) and Figure 4 (line 232).

8) “(…) 25.45 ± 2.43 min, in contrast to (…) 57.03 ± 9.28 min. (…) E. coli stimulation was 8.7±3.3 min and for S. aureaus 5.4 ± 4.6 min” (lines 282-286). Please, the authors should homogenize the significant figures. Same comment for Table 2 (line 326).

9) Discussion. Please, the authors should highlight some potential avenues that this current research could pursue in the near future.

OVERVIEW AND FINAL COMMENTS

The submitted work is well-designed and the gathered results are interesting to have a more complete outlook of the neutrophil granulocytes performance under the presence of reactive oxygen species. This fact will aid to design more efficient drug therapies against inflammatory and immune diseases. For these reasons, I will recommend the present scientific manuscript for further publication in Biomedicines once all the aforementioned suggestions will be properly fixed.

The scientific paper is well written in general terms. Neverthelss, the authors should recheck the manuscript in order to improve final details.

Author Response

Dear colleague, thank you very much for your work. We have fixed issues according to your comments.

The manuscript titled “ROS production by single cell and populations of neutrophils upon bacterial stimulation” by Pleskova, S.N.; et al. is an original work where the authors address the impact of the presence of two different bacteria (Staphylococcus aureus and Escherichia coli) in the generation of reactive oxygen species (ROS) in neutrophil granulocyte bodies by scanning ion-conductance microscopy (SICM). The authors found a significantly larger amount of ROS production when the neutrophils were exposed to Staphylococcus aureus in comparison to Escherichia coli.

The most relevant outcomes found by the authors can contribute to better understand the underlying redox mechanisms involved inside of neutrophil granulocytes. This knowledge could have a positive impact to design more efficient anti-inflammatory treatments against diseases caused by bacteria. Furthermore, the methodology followed by the authors can be fully extandable to other immune cells.

Here, there exists some suggestions in order to improve the scientific quality of the manuscript paper:

1) Introduction section is clear and concise. “(…) since neutrophil granulocytes (NGs) (…) play the regulatory role as a conductor in the inflammation process” (lines 34-36). Here, the authors should add some relevant reference citation [1].

[1] Domerecka, W.; et al. Indicator of Inflammation and NETosis-Low-Density Granulocytes as a Biomarker of Autoimmune Hepatitis. J. Clin. Med. 2022, 11, 2174. https://doi.org/10.3390/jcm11082174

Thank you for the note, the citation was added.

2) “the following subpopulations: (…) cells with high chemotactic activity (…) (about 60%), (2) cells that did not show any chemotactic activity (…) (about 28%), and (3) cells with high-motor activity in the cytoplasm and not capable of quantum dot phagocytosis (about 10 %)” (lines 49-54). Why does the sum of all this percentages not render a total of 100%? Please, the authors should fix this point.

Thank you for this note, we checked information in the original source and rounded up the numbers to required 100%.

3) “(…) neutrophils were divided into two subpopulations: (…) NBT (Nk)” (lines 62-65). Here, the authors should add the percentages according each subpopulation in order to be consistent with the previous information detailed in this section.

Thank you for the note, the percentage was added.

4) “The release of ROS/RNS (…) was demonstrated by electrochemical methods” (lines 91-93). Here, the authors show a suitable approach to determine the impact of ROS and RNS inside the phagolysosome cells but it exist many others. For example, the authors should also point out the possibility to use nanoindentation techniques [2] to unravel the effect of ROS/RNS species inside the cell like the recently reported study where the cellular elasticity properties were monitored according the presence of ROS [3]. Finally, the authors should highlight the advantages of those measurements carried out by scanning ion-conductance microscopy like the fact not to touch the sample during scanning which greatly benefits not to damage the cellular samples of interest.

[2] Magazzù, A.; et al. Investigation of Soft Matter Nanomechanics by Atomic Force Microscopy and Optical Tweezers: A Comprehensive Review. Nanomaterials 2023, 13, 963. https://doi.org/10.3390/nano13060963.

[3] Pastrana, H.F.; et al. Evaluation of the elastic Young’s modulus and cytotoxicity variations in fibroblasts exposed to carbon-based nanomaterials. J. Nanobiotechnology. 2019, 17, 32. https://doi.org/10.1186/s12951-019-0460-8.

Thank you for the note. We added the references to the text, but in this modification of the SICM method we penetrate cells by injection of nanoelectrod. You are absolutely right that SICM method allows non-contact study of cell morphology (unlike to AFM even in resonance mode) but here we utilized different method. Anyway despite the injection the physiology of cell is not violated.

5) Materials and Methods. Please, the authors should detail all manufacturer details (name and country) for all used chemical agents and consumables. For example, there lacks this information related to employed buffers (PBS, NaCl and KCl in lines 109-110 or GRM agar in line 117). Please, take this comment into account for the rest of the M&M section.

Thank you, information was added.

6) (OPTIONAL) Perhaps it may be interesting to add a extra Figure with some results according to the Fabrication and calibration of nanoelectrodes (lines 122-137) as Supplementary Information (SI) in order to see the subsequent oxidation of ferrocenemethanol molecules and platinum deposition in each consecutive cycle.

We have previously described the fabrication and calibration of this sensor, so we have provided a reference to the relevant research in the text.

7) Results and Discussion. Figure 1, panel B (line 192) and Figure 2 (line 197). Are the axes of these chronoamperograms (with and without stimulation respectively) the same range of values? Same comment for the X-axis of Fig. 3, panel C respect the respective panels A and B (line 214) and Figure 4 (line 232).

Thanks for the comment. Figure 1 shows a schematic drawing, this is a clipping from a chronoamperogram. Next, we aligned all the graphs (Figure 2 and 3) within the X-axis in accordance with your comment.      

8) “(…) 25.45 ± 2.43 min, in contrast to (…) 57.03 ± 9.28 min. (…) E. coli stimulation was 8.7±3.3 min and for S. aureaus 5.4 ± 4.6 min” (lines 282-286). Please, the authors should homogenize the significant figures. Same comment for Table 2 (line 326).

Thank you for the note, numbers were homogenized.

9) Discussion. Please, the authors should highlight some potential avenues that this current research could pursue in the near future.

Thank you for the suggestion, we highlighted some moments at the end of the discussion.

OVERVIEW AND FINAL COMMENTS

The submitted work is well-designed and the gathered results are interesting to have a more complete outlook of the neutrophil granulocytes performance under the presence of reactive oxygen species. This fact will aid to design more efficient drug therapies against inflammatory and immune diseases. For these reasons, I will recommend the present scientific manuscript for further publication in Biomedicines once all the aforementioned suggestions will be properly fixed.

Reviewer 3 Report

The data presented in this MS suggest that normal granulocytes have a higher ROS-mediated reactivity to S. aureus than to E. coli.

My comments and suggestions.

Major:

1.       The MS is poorly structured and extremely hard to read. The “Results and Discussion” section should be divided into respective two sections, and the “Results” section into at least two subsections.

2.       Almost the entire “Introduction” section is devoted to the morphological and functional heterogeneity of neutrophils. However, the MS focuses on the selective reactivity of granulocytes to different bacteria, and not neutrophil heterogeneity. The mechanisms of neutrophil reactivity were also completely left out of the discussion: the authors discuss only methodological details, not the essence of a biological phenomenon.

3.       Strictly speaking, the authors examined the reactivity of granulocytes, which include not only neutrophils but also eosinophils and basophils. This must be reflected in the MS.

 Minor:

The functional difference between single neutrophils and a neutrophil population is not entirely clear from the MS. Instead of the term “ROS production by single neutrophils”, it would be better to use the term “intracellular ROS production by neutrophils”.

Author Response

Dear colleague, thank you very much for your work. We have fixed issues according to your comments.

The data presented in this MS suggest that normal granulocytes have a higher ROS-mediated reactivity to S. aureus than to E. coli.

My comments and suggestions.

Major:

  1. The MS is poorly structured and extremely hard to read. The “Results and Discussion” section should be divided into respective two sections, and the “Results” section into at least two subsections.

Thank you for the note, we revised the structure of the manuscript.

  1. Almost the entire “Introduction” section is devoted to the morphological and functional heterogeneity of neutrophils. However, the MS focuses on the selective reactivity of granulocytes to different bacteria, and not neutrophil heterogeneity. The mechanisms of neutrophil reactivity were also completely left out of the discussion: the authors discuss only methodological details, not the essence of a biological phenomenon.

Thank you. The main idea of the article was to highlight the heterogeneity in the reaction of different subpopulation of neutrophils to various bacterial strains. But it’s possible only using single cell analyzes methods. So the novelty of the MS is the using of brand new method of amperometric nanoelectrode measurements inside the living cell. Whereas it was important to discuss methodological details. Also we discussed the patterns of cells behavior in single cell level and population level, we hope it became more clear after the dividing of results and discussion.

  1. Strictly speaking, the authors examined the reactivity of granulocytes, which include not only neutrophils but also eosinophils and basophils. This must be reflected in the MS.

Thank you for your note, but actually we analyzed the neutrophil population using flow cytometery and it showed that population purity was 98-99%. We added this information to the materials and methods. Also the saw the typical morphology of neutrophil using microsopy.

Minor:

The functional difference between single neutrophils and a neutrophil population is not entirely clear from the MS. Instead of the term “ROS production by single neutrophils”, it would be better to use the term “intracellular ROS production by neutrophils”.

Thank you very much, we completely agree with you and changed this term.

Round 2

Reviewer 1 Report

None

Author Response

Dear colleague, thank you very much for your work. We have fixed issues according to your comments and significantly edited the text of the manuscript.

Reviewer 3 Report

The authors have significantly improved the article, but not enough.

 1. The title of the article is not entirely clear. The title might be better, for example: ROS production by a single neutrophil cell and neutrophil population upon bacterial stimulation.

2. The data show a higher reactivity of neutrophils to S. aureus compared to E. coli.  This phenomenon needs, at least, a tentative explanation (TLR ligand density?, soluble bacterial products ?…) but there is no such explanation.

3. The text is hard to read and needs to be edited.

Author Response

Independent Review Report, Reviewer 3

Dear colleague, thank you very much for your work. We have fixed issues according to your comments and significantly edited the text of the manuscript.

The authors have significantly improved the article, but not enough.

  1. The title of the article is not entirely clear. The title might be better, for example: ROS production by a single neutrophil cell and neutrophil population upon bacterial stimulation.

            Thanks for the comment. We corrected the title.

  1. The data show a higher reactivity of neutrophils to S. aureus compared to E. coli. This phenomenon needs, at least, a tentative explanation (TLR ligand density?, soluble bacterial products ?…) but there is no such explanation.

            We have added explanation to the manuscript (line 1357-1369).

  1. The text is hard to read and needs to be edited.

Thank you for your comment. We have carefully edited the text and improved the English language.We have greatly simplified the text for reading.
